# Challenges of early renal cancer detection: symptom patterns and incidental diagnosis rate in a multicentre prospective UK cohort of patients presenting with suspected renal cancer

Naveen S Vasudev ![ORCID],[1] Michelle Wilson,[1] Grant D Stewart,[2,3] Adebanji Adeyoju,[4] Jon Cartledge,[5] Michael Kimuli,[5] Shibendra Datta,[6] Damian Hanbury,[7] David Hrouda,[8] Grenville Oades,[9] Poulam Patel,[10] Naeem Soomro,[11] Mark Sullivan,[12] Jeff Webster,[13] Peter J Selby,[1] Rosamonde E Banks[1]

For numbered affiliations see end of article.

**Correspondence to**
Dr Naveen S Vasudev;
n.vasudev@leeds.ac.uk

## ABSTRACT

**Objectives** To describe the frequency and nature of symptoms in patients presenting with suspected renal cell carcinoma (RCC) and examine their reliability in achieving early diagnosis.

**Design** Multicentre prospective observational cohort study.

**Setting and participants** Eleven UK centres recruiting patients presenting with suspected newly diagnosed RCC. Symptoms reported by patients were recorded and reviewed. Comprehensive clinico-pathological and outcome data were also collected.

**Outcomes** Type and frequency of reported symptoms, incidental diagnosis rate, metastasis-free survival and cancer-specific survival.

**Results** Of 706 patients recruited between 2011 and 2014, 608 patients with a confirmed RCC formed the primary study population. The majority (60%) of patients were diagnosed incidentally. 87% of patients with stage Ia and 36% with stage III or IV disease presented incidentally. Visible haematuria was reported in 23% of patients and was commonly associated with advanced disease (49% had stage III or IV disease). Symptomatic presentation was associated with poorer outcomes, likely reflecting the presence of higher stage disease. Symptom patterns among the 54 patients subsequently found to have a benign renal mass were similar to those with a confirmed RCC.

**Conclusions** Raising public awareness of RCC-related symptoms as a strategy to improve early detection rates is limited by the fact that related symptoms are relatively uncommon and often associated with advanced disease. Greater attention must be paid to the feasibility of screening strategies and the identification of circulating diagnostic biomarkers.

## INTRODUCTION

The incidence of kidney cancer in Europe is among the highest worldwide. In the UK,

### Strengths and limitations of this study

► The multicentre, prospective nature of this study, among a contemporary cohort of UK patients, is unique and represents an important strength over previous studies.

► Comprehensive linked clinico-pathological and outcome data were available for all patients.

► Symptoms among patients subsequently found to have a benign renal mass are reported in parallel.

► This was not a population-based study and our cohort represents only a small proportion of all patients diagnosed with renal cell carcinoma in the UK within the study period.

► Patient-reported symptoms were recorded following referral to secondary care and may therefore be subject to recall bias.

incidence rates have risen by 47% over the past decade, with 12 000 new cases in 2015.[1] By 2035, it is predicted that this number will rise to over 20 000 new cases per annum and kidney cancer will come to represent the fourth most common cancer among men and ninth most common among women in the UK.[2]

Diagnosing patients with kidney cancer can be challenging.[3] Renal cell carcinomas (RCCs), which make up the majority (85%) of kidney cancers, are characteristically insidious in onset. The once classical triad of haematuria, pain and abdominal mass is now recognised to be rare, and symptoms, if present at all, can be vague, non-specific and delayed in onset. While early diagnosis is recognised to be key in achieving optimal

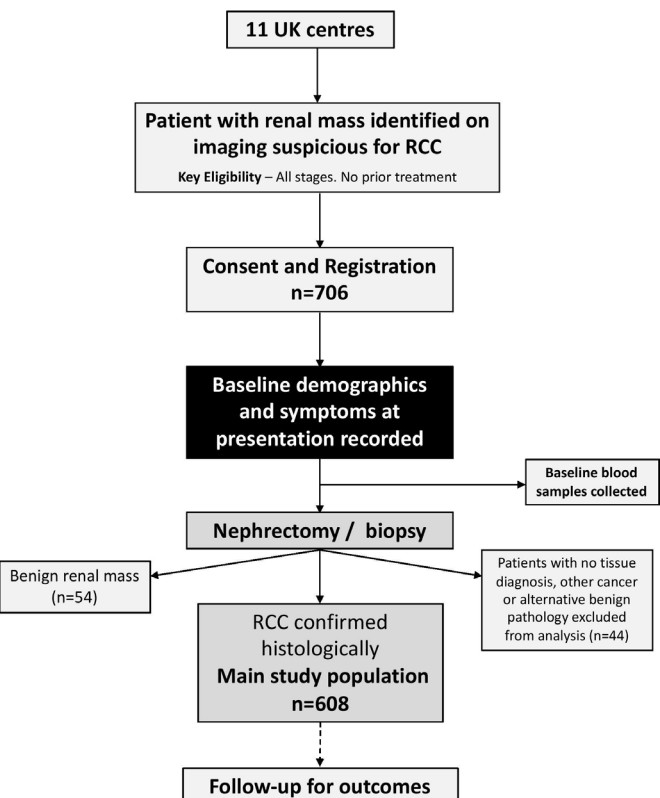

11 UK centres

Patient with renal mass identified on imaging suspicious for RCC
**Key Eligibility** – All stages. No prior treatment

Consent and Registration
n=706

Baseline demographics and symptoms at presentation recorded

Baseline blood samples collected

Nephrectomy / biopsy

Benign renal mass (n=54)

Patients with no tissue diagnosis, other cancer or alternative benign pathology excluded from analysis (n=44)

RCC confirmed histologically
**Main study population**
n=608

Follow-up for outcomes

**Figure 1** Flowchart of patients through the study. RCC, renal cell carcinoma.

outcomes, many patients still present with advanced disease. In 2017 in England, for example, figures show that among patients with a recorded stage at diagnosis, 19% had stage III and 23% had stage IV disease, at the time of presentation.[4]

Campaigns to raise awareness of kidney cancer among the public and doctors have been employed in an effort to improve early diagnosis rates.[5] Understanding how patients present may help to inform such strategies. Unlike previous studies, we prospectively collected information on symptoms reported by patients at the time of their diagnosis of suspected RCC, following recruitment to a large, contemporary, multi-institutional UK RCC biobank.[6] The aims of this substudy were to describe symptoms reported by patients, define the current rate of incidental diagnosis and look at how these factors relate to patient outcomes, with the goal of better understanding the challenges in early RCC diagnosis.

## METHODS
The design was a multicentre prospective observational cohort study. Patients with a renal mass on imaging suspicious of RCC, of all stages, with no prior treatment, were eligible. Patients were approached and consented to participate in the study prior to surgery or biopsy, before diagnosis of RCC was confirmed. Full details regarding the inclusion and exclusion criteria are as previously

reported.[6] Comprehensive clinical and pathological information was collected.

At the time of recruitment to the study, patients were asked about the presence and nature of symptoms leading to their diagnosis of suspected RCC, which was recorded using paper case report forms (CRF). Specific information relating to commonly related 'RCC-type' local symptoms (pain, haematuria, abdominal mass and/or other) and systemic symptoms (weight loss (any), loss of appetite, sweats, fevers, fatigue and/or other) was recorded. In addition, the investigator completing the CRF was asked to state whether the diagnosis was incidental in nature and included a subsequent free-text box requesting a description of how the patient was diagnosed. All cases were independently reviewed by two reviewers (NSV and REB) to confirm or refute whether the diagnosis would be regarded as incidental or not (ie, were any symptoms reported and, if so, would they be regarded as being related to the finding of RCC), with additional reference to individual electronic case notes where available. The reported presence of RCC-type symptoms, many of which, such as pain, are non-specific, was not always related to the finding of RCC and, where applicable, therefore, considered incidental. Cases with insufficient data or where the incidental nature of the diagnosis remained uncertain were not classified. Patients being investigated for asymptomatic hypertension were not classified as incidental.[7]

Metastasis-free survival (MFS) was calculated for patients with localised disease, defined as the period from the date of nephrectomy to the date of distant recurrence. Patients without recurrence were censored at the date they were last known to be recurrence-free (for patients who died without recurrence this was the date of death). Cancer-specific survival (CSS) was defined as the period from the date of nephrectomy to the date of cancer-related death. Patients with a non-cancer related death were censored at their date of death and patients still alive were censored at the last date they were known to be alive. Kaplan-Meier plots were produced to visualise survival and the log-rank test was used to detect a statistically significant difference between survival curves.

### Public and patient involvement
Patients were extensively involved in the design, delivery and evaluation of the NIHR Programme supporting this work. Patients were not directly involved in the design or evaluation of the current report.

## RESULTS
Between July 2011 and June 2014, 706 patients were recruited to the study from 11 UK centres (8 England, 2 Scotland, 1 Wales). Details regarding recruitment by centre are shown in online supplementary table 1. The flow of patients through the study is shown in figure 1. RCC was confirmed in 608 (86%) patients, among whom median follow-up was 4.8 years (IQR: 3.7, 5.2), and benign renal mass in 54 (7.6%) patients. The

remaining 44 (6.4%) patients either did not undergo biopsy or nephrectomy or had no tumour in their biopsy cores (n=33), had another (not RCC) malignancy (n=5) or an alternative benign pathology (n=6).[6] Among all patients with a confirmed RCC, 422 (69%) patients reported having RCC-type symptoms at diagnosis, of whom 221 (52%) reported symptoms that were considered related to the presence of RCC. Among these 221

**Table 1** Patient and tumour characteristics by symptom type for continuous variables, figures in table represent median (range) with corresponding p value from the Kruskal-Wallis test and for categorical variables, figures in table represent n (%) with corresponding p value from the $\chi^2$ test

| Characteristic | No RCC-type symptoms (n=186) | RCC-type symptoms reported (n=422)† | | | P value |
| --- | --- | --- | --- | --- | --- |
| | | Not RCC related (n=183) | RCC-related local symptoms only (n=97) | RCC-related systemic symptoms (±local) (n=124) | |
| Age (years) | 65 (31–86) | 63 (29–90) | 63 (38–84) | 62 (33–92) | 0.31 |
| Gender | | | | | |
| Female | 67 (32.7) | 62 (30.2) | 21 (10.2) | 55 (26.8) | |
| Male | 119 (30.9) | 121 (31.4) | 76 (19.7) | 69 (17.9) | 0.01 |
| BMI | 28.5 (15.6–74.4) | 27 (18.1–56.5) | 28.8 (17.3–67.2) | 27.5 (16–54.5) | 0.01 |
| Tumour size (mm) | 44 (14–180) | 43 (11–170) | 74 (13–155) | 75 (20–240) | <0.01 |
| pT | | | | | |
| 1a | 83 (42.6) | 88 (45.1) | 16 (8.2) | 8 (4.1) | |
| 1b | 46 (34.3) | 42 (31.3) | 19 (14.2) | 27 (20.1) | |
| 2 | 15 (19.7) | 18 (23.7) | 19 (25) | 24 (31.6) | |
| 3 | 38 (22.6) | 33 (19.6) | 42 (25) | 55 (32.7) | |
| 4 | 0 (0) | 0 (0) | 1 (25) | 3 (75) | |
| X | 1 (50) | 1 (50) | 0 (0) | 0 (0) | |
| Missing | 1 (100) | 0 (0) | 0 (0) | 0 (0) | |
| NA | 2 (20) | 1 (10) | 0 (0) | 7 (70) | <0.01 |
| Grade | | | | | |
| 1 | 4 (40) | 0 (0) | 4 (40) | 2 (20) | |
| 2 | 55 (34.8) | 50 (31.6) | 25 (15.8) | 28 (17.7) | |
| 3 | 88 (32.2) | 94 (34.4) | 47 (17.2) | 44 (16.1) | |
| 4 | 13 (14.9) | 13 (14.9) | 19 (21.8) | 42 (48.3) | |
| Missing | 9 (39.1) | 9 (39.1) | 1 (4.3) | 4 (17.4) | |
| NA | 17 (43.6) | 17 (43.6) | 1 (2.6) | 4 (10.3) | <0.01 |
| Stage | | | | | |
| I | 130 (39.8) | 129 (39.4) | 34 (10.4) | 34 (10.4) | |
| II | 12 (17.4) | 17 (24.6) | 18 (26.1) | 22 (31.9) | |
| III | 34 (24.5) | 29 (20.9) | 37 (26.6) | 39 (28.1) | |
| IV | 10 (18.9) | 6 (11.3) | 8 (15.1) | 29 (54.7) | |
| Missing | 0 (0) | 2 (100) | 0 (0) | 0 (0) | <0.01 |
| Tumour subtype | | | | | |
| Clear cell | 147 (31.7) | 137 (29.6) | 83 (17.9) | 96 (20.7) | |
| Papillary | 16 (27.1) | 23 (39) | 7 (11.9) | 13 (22) | |
| Chromophobe | 15 (32.6) | 15 (30.4) | 7 (15.2) | 10 (21.7) | |
| Unclassified | 7 (38.9) | 6 (33.3) | 0 (0) | 5 (27.8) | |
| Other | 1 (33) | 2 (67) | 0 (0) | 0 (0) | 0.81 |

*Not applicable (NA)—patients underwent biopsy only or tumour ablation.
†18 patients reported symptoms but their relationship to RCC could not be determined.
BMI, body mass index; RCC, renal cell carcinoma.

**Table 2** Patient and tumour characteristics by diagnosis type: for continuous variables, figures in table represent median (range) with corresponding p-value from the Wilcoxon rank-sum test, and for categorical variables, figures in table represent n (%) with corresponding p-value from the $\chi^2$ test

| Characteristic | Non-incidental (n=231) | Incidental (n=351) | P value |
|---|---|---|---|
| Age (years) | 62 (33–92) | 65 (29–90) | 0.04 |
| Gender | | | |
| Female | 77 (38.3) | 124 (61.7) | |
| Male | 154 (40.4) | 227 (59.6) | 0.69 |
| BMI | 28.3 (15.6–67.2) | 27.8 (17.2–57.7) | 0.38 |
| Tumour size (path) (mm) | 75 (13–240) | 42 (11–170) | <0.01 |
| Tumour size (CT) (mm) | 80 (16–250) | 44 (10–170) | <0.01 |
| pT | | | |
| 1a | 25 (12.8) | 170 (87.2) | |
| 1b | 48 (37.2) | 81 (62.8) | |
| 2 | 46 (60.5) | 30 (39.5) | |
| 3 | 101 (61.2) | 64 (38.8) | |
| 4 | 4 (100) | 0 (0) | |
| X | 0 (0) | 2 (100) | |
| Missing | 0 (0) | 1 (100) | |
| NA* | 7 (70) | 3 (30) | <0.01 |
| Grade | | | |
| 1 | 6 (66.7) | 3 (33.3) | |
| 2 | 56 (35.9) | 100 (64.1) | |
| 3 | 93 (34.6) | 176 (65.4) | |
| 4 | 65 (75.6) | 21 (24.4) | |
| Missing | 6 (26.1) | 17 (73.9) | |
| NA* | 5 (12.8) | 34 (87.2) | <0.01 |
| Stage | | | |
| I | 70 (21.6) | 254 (78.4) | |
| II | 42 (60.9) | 27 (39.1) | |
| III | 80 (58.4) | 57 (41.6) | |
| IV | 39 (78) | 11 (22) | |
| Missing | 0 (0) | 2 (100) | <0.01 |
| Tumour subtype | | | |
| Clear cell | 186 (40.9) | 269 (59.1) | |
| Papillary | 21 (35.6) | 38 (64.4) | |
| Chromophobe | 19 (41.3) | 28 (58.7) | |
| Unclassified | 5 (27.8) | 13 (72.2) | |
| Other | 0 (0) | 3 (100) | 0.62 |

*Not applicable (NA), patients underwent biopsy only or tumour ablation.
BMI, body mass index.

patients, 97 (44%) had local symptoms only, 19 (8.6%) had systemic symptoms only and 105 (47.5%) reported having both local and systemic symptoms. Patient and tumour characteristics by symptom type are shown in table 1.

**Table 3** Nature of incidental diagnosis

| Type of incidental diagnosis | n (%) |
|---|---|
| Investigation for pre-existing condition | 65 (18) |
| Another malignancy | 34 (53) |
| Diabetes mellitus | 7 (11) |
| Hepatobiliary* | 5 (8) |
| AAA screening/post-aortic repair | 3 (5) |
| Other† | 16 (23) |
| Investigation for signs or symptoms unrelated to RCC | 258 (74) |
| Gastrointestinal‡ | 86 (33) |
| Urinary tract§ | 49 (19) |
| Hepatobiliary¶ | 27 (10) |
| Respiratory** | 20 (8) |
| Musculoskeletal†† | 16 (6) |
| Cardiovascular‡‡ | 11 (4) |
| Trauma | 7 (3) |
| Gynaecological | 6 (3) |
| Anaemia | 4 (2) |
| Miscellaneous§§ | 32 (12) |
| Routine health check¶¶ | 16 (5) |
| Not known*** | 12 (3) |

*Cirrhosis, primary biliary cirrhosis and sclerosing cholangitis.
†Includes Addison's disease, chronic renal failure, Crohn's disease, coeliac disease, ovarian cyst, renal stones, IgA nephropathy, Wegener's granulomatosis, polymyalgia rheumatica and ovarian cyst.
‡Altered bowel habit, GI bleed, bloating/distension, abdominal pain and reflux.
§Urinary retention, prostatic symptoms, high prostate-specific antigen, urosepsis, renal colic and impaired renal function.
¶Biliary colic, deranged liver function tests, jaundice, pancreatitis and cholecystitis.
**Shortness of breath, cough, haemoptysis and pneumonia.
††Back pain, leg pain and joint pain.
‡‡Chest pain, myocardial infarction, claudication and endocarditis.
§§Includes dizziness, syncope, elevated blood test values and ankle swelling.
¶¶Initial investigations were urine dip (6), ultrasound scan (5), CT scan (2), blood tests (2) and chest x-ray (1).
***Insufficient information to classify.
AAA, abdominal aortic aneurysm; RCC, renal cell carcinoma.

## Local RCC-related symptoms

Among the 202 (33%) patients reporting local RCC-related symptoms, 137 (68%) reported visible haematuria and 126 (62%) reported pain, with only 14 (7%) patients reporting an abdominal mass. Patients presenting with haematuria had a median pathological tumour size of 75 mm (range 16–155) and almost half had stage III (37.2%) or IV (12.4%) disease. Only four patients (0.6%) presented with the classical triad of an abdominal mass, haematuria and local pain. The median tumour size among these four patients was 105 mm (range 80–154 mm) on preoperative cross-sectional imaging. No significant differences were present when considered by histological type, although the small number of patients with non-clear cell RCC limits this comparison.

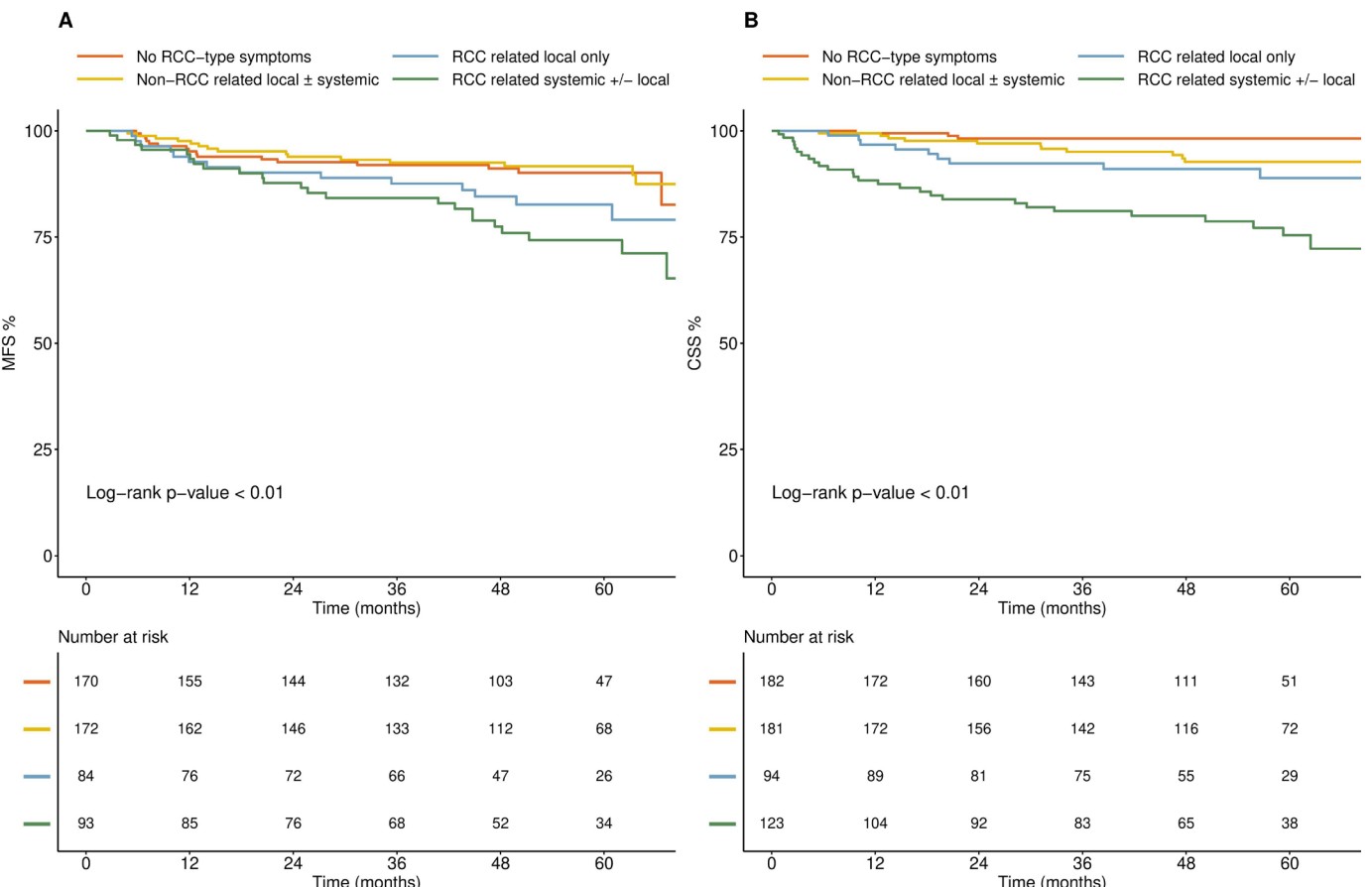

**Figure 2** Kaplan Meier survival curves by symptom type. Survival outcomes ((A) MFS; (B) CSS) in patients with no RCC-type symptoms, unrelated RCC-type symptoms, local RCC-related symptoms and those with systemic (± local) RCC-related symptoms. CSS, cancer-specific survival; MFS, metastasis-free survival; RCC, renal cell carcinoma.

### Systemic RCC-related symptoms

Among those reporting systemic symptoms related to their RCC, fatigue (62%), weight loss (52%), sweats (38%) and loss of appetite (38%) were all commonly reported. Fever was relatively uncommon (10%). Patients with systemic symptoms were more likely to have grade 4 cancers and stage IV disease than those with local RCC-related symptoms only and those with symptoms unrelated to RCC (p<0.01) (table 1).

### Incidental diagnosis

Among the 582 patients in whom the nature of the diagnosis could be confidently classified, 351 (60%) cases of RCC were deemed to have been diagnosed incidentally. Patient and tumour characteristics by nature of diagnosis (incidental vs non-incidental) are shown in table 2. No association with patient sex was found and distribution of histological subtype was similar between groups. Non-incidentally detected tumours were larger and of higher grade and stage than incidentally detected tumours (p<0.01). Among patients diagnosed with a localised pT1a tumour, the incidental diagnosis rate was 87%. Conversely, 22% of patients with stage IV disease were considered to have been diagnosed incidentally. The nature of the incidental diagnosis (eg, during investigation for a known pre-existing condition vs investigation of unrelated symptoms) is shown in table 3.

### Tumour size

Pathological tumour size was available for 556 (91%) of patients. We looked at symptoms in patients presenting with tumours ≥10 cm. Among the 66 patients with a tumour ≥10 cm, 31 (47%) reported haematuria at the time of presentation, 33 (50%) reported pain and abdominal mass was reported in four (6%) patients. Almost a quarter (16/66; 24%) of these patients were considered to have been diagnosed incidentally, with 10 (15%) reporting no symptoms, despite the presence of a large primary tumour. No effect of BMI was observed in relation to presence or absence of symptoms.

### Outcomes

We looked at survival outcomes by both symptom type (no RCC-type symptoms or unrelated RCC-type symptoms vs related RCC-type symptoms) and incidental vs non-incidental diagnosis. Patients diagnosed with no RCC-type symptoms and those reporting unrelated RCC-type symptoms had a significantly improved MFS and CSS compared with patients with related RCC-type symptoms. Furthermore, patients with systemic RCC-related

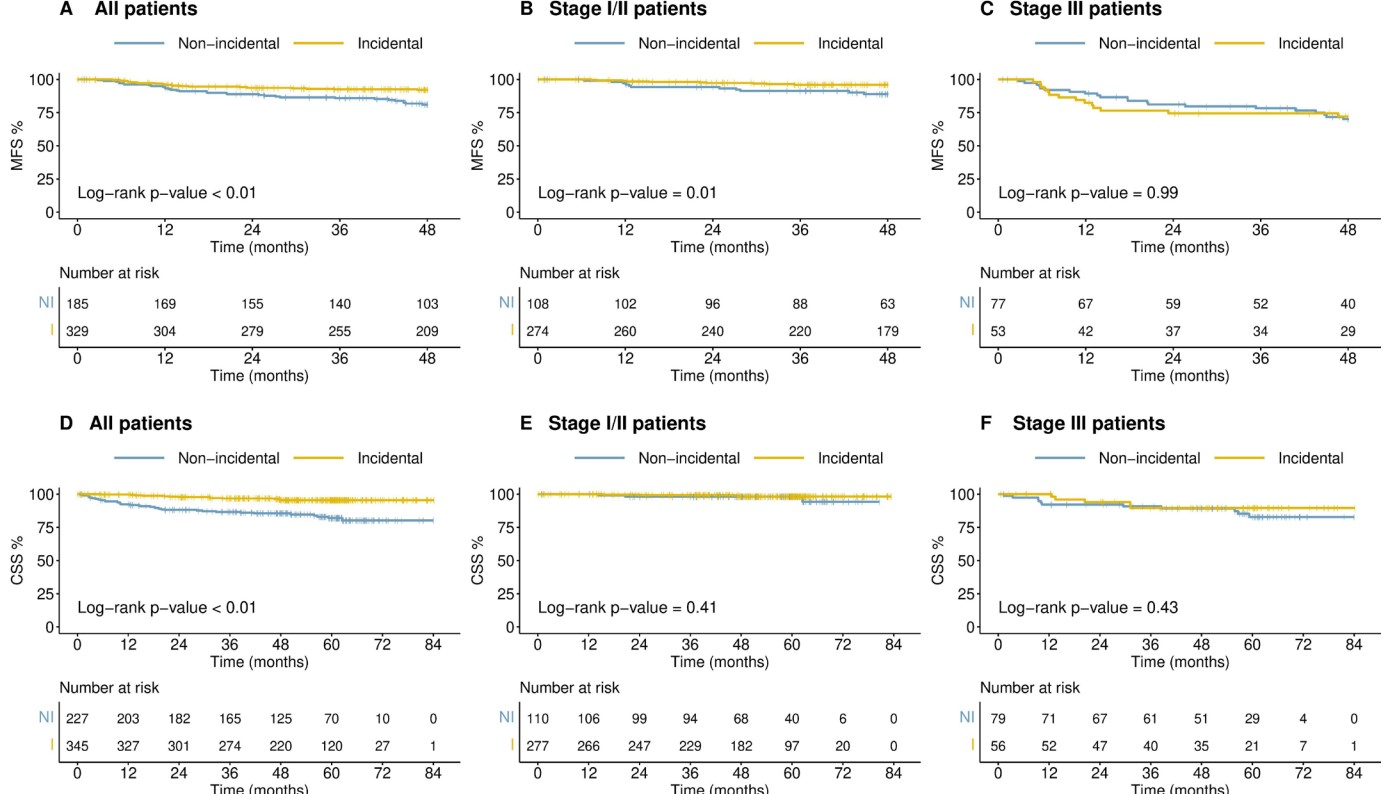

**Figure 3** Kaplan-Meier survival curves by incidental versus non-incidental diagnosis for all patients, stage I/II or stage III RCC. (A–C) MFS; (D–F) CSS. CSS, cancer-specific survival; MFS; metastasis-free survival; RCC, renal cell carcinoma.

symptoms had poorer outcomes than those with local RCC symptoms only (figure 2A,B). Overall, patients with an incidental diagnosis of RCC had improved MFS and CSS in comparison to those diagnosed non-incidentally, although it is important to note that these effects were lost when controlled for stage of disease (figure 3).

### Patients presenting with benign renal masses
In total, 54 (7.6%) patients in our cohort were found to have a benign renal mass, composed of oncocytoma (n=29), angiomyolipoma (n=8) and other lesions (n=17) (table 4). The incidental diagnosis rate was 56% among the 52 evaluable patients. Haematuria and pain were reported, respectively, in 57% and 52% of patients diagnosed non-incidentally. The majority (65%) reported symptoms, of whom 57% had local symptoms only, 17% had systemic symptoms only and 26% reported both local and systemic symptoms.

### DISCUSSION
Early detection is widely held to be a key strategy towards improving outcomes in patients with RCC.[8] As in most solid cancers, disease stage and survival are closely linked, with 3 year CSS rates in our cohort, for example, of 99% and 47% for stage I and stage IV cancers, respectively (data not shown). Symptoms of kidney cancer such as visible haematuria and flank pain are well documented and NHS initiatives such as 'be clear on cancer: blood in your pee' campaign have been aimed at prompting

the public to seek early medical attention.[5] Nevertheless, many patients still present with overt or micro-metastatic disease. Understanding the type and frequency of symptoms patients with newly diagnosed RCC report is critical in beginning to address this issue and understand whether simply raising awareness among doctors and the public is sufficient or other strategies are needed.

Our study highlights the significant challenges in diagnosing patients with kidney cancer. Almost a third of patients in our cohort were symptomless at the time of diagnosis, among whom nearly a quarter (24%) had stage III or IV disease. Visible haematuria, a hallmark symptom of this disease, was recorded in just 23% of patients overall. Even among patients with large (≥10 cm) tumours, less than half (47%) reported haematuria as a symptom. Prior reports using UK general practice database records have suggested rates of haematuria as low as 18% in patients presenting with kidney cancer, compounded by the low positive predictive value (PPV) (1%) of this symptom for RCC among those ≥60 year old.[9] Furthermore, symptom patterns do not appear to reliably distinguish patients with benign renal masses from those with RCC.

Many studies have attempted to document the incidental diagnosis rate for renal cancer. These previous studies have all been retrospective in nature, typically derived from patients at a single centre, with widely varying rates of incidental diagnosis, from 15% to 61%, in a less contemporaneous setting (broadly spanning 1970–2000).[10–14] A more recent, global, study, involving

**Table 4** Characteristics and symptoms associated with benign renal masses

| Characteristic | All (n=54) | Oncocytoma (n=29) | AML (n=8) | Other* (n=17) |
|---|---|---|---|---|
| Age (years) | 65 (32–86) | 66 (42–86) | 63 (59–68) | 61 (32–78) |
| Gender | | | | |
| Female | 29 (53.7) | 12 (41.4) | 5 (62.5) | 12 (70.6) |
| Male | 25 (46.3) | 17 (58.6) | 3 (37.5) | 5 (29.4) |
| BMI | 27.6 (18.7–45.8) | 27.8 (19.4–39.6) | 28 (22–38.8) | 26.4 (18.7–45.8) |
| CT size (cm) | | | | |
| ≤4 | 22 (44.9) | 14 (50) | 3 (50) | 5 (33.3) |
| 4< to ≤7 | 18 (36.7) | 11 (39.3) | 2 (33.3) | 5 (33.3) |
| 7< to ≤10 | 6 (12.2) | 1 (3.6) | 1 (16.7) | 4 (26.7) |
| >10 | 3 (6.1) | 2 (7.1) | 0 (0) | 1 (6.7) |
| NA | 5 (-) | 1 (-) | 2 (-) | 2 (-) |
| RCC-type symptoms | | | | |
| No | 19 (35.2) | 10 (34.5) | 4 (50) | 5 (29.4) |
| Yes | 35 (64.8) | 19 (65.5) | 4 (50) | 12 (70.6) |
| Local symptoms | | | | |
| No | 6 (17.1) | 3 (15.8) | 1 (25) | 2 (16.7) |
| Yes | 29 (82.9) | 16 (84.2) | 3 (75) | 10 (83.3) |
| Systemic symptoms | | | | |
| No | 20 (57.1) | 12 (63.2) | 1 (25) | 7 (58.3) |
| Yes | 15 (42.9) | 7 (36.8) | 3 (75) | 5 (41.7) |
| Incidental diagnosis | | | | |
| No | 23 (42.5) | 13 (44.8) | 2 (25) | 8 (47) |
| Yes | 29 (54) | 15 (51.7) | 6 (75) | 8 (47) |
| Not known | 2 (3.5) | 1 (3.5) | 0 (0) | 1 (6) |

*Consists of cystic nephroma (4), benign cyst (3), metanephric adenoma (2), mixed epithelial stromal tumour (2), haemangioblastoma (1), leiomyomata (1), multilocular cyst (1), myxoid mesenchymal tumour (1), Rosai Dorfman disease (1), and solitary fibrous tumour (1).
AML, angiomyolipoma; BMI, body mass index; RCC, renal cell carcinoma.

4288 patients presenting with RCC between 2010 and 2012, reported an incidental diagnosis rate of 67%.[15] However, no detail regarding how this was derived, or the nature and characteristics of those diagnosed incidentally, were presented in this study. While retrospective studies have the advantage of being feasible on a large scale, often with long-term follow-up data, recording of symptoms at presentation may not have been performed for this purpose and may, therefore, not be complete. Furthermore, determining whether a diagnosis is incidental or not can often require further detail beyond the recording of symptoms alone, and which may not always be available when records are reviewed retrospectively. Here, we collected symptoms reported by patients at diagnosis in a planned way as part of the study design using standardised CRFs, allowed for detailed free-text annotation of the history leading to the diagnosis and asked investigators to specifically indicate whether this was felt to be incidental in nature. We carefully reviewed the presenting symptoms and history for each patient in our study, performed independently by two of the authors, to determine as accurately as possible whether the diagnosis would be deemed incidental or not. Pain, for example,

was a commonly reported symptom not necessarily attributable to the diagnosis of RCC, for example, when located in an anatomically distinct site. We believe our figure of 60%, among a contemporary set of patients (2011–2014), provides a true reflection of the current incidental diagnosis rate of RCC in the UK, and supports the general rise in the incidental detection of kidney cancer that has been reported over time.

Our data show that the majority (60%) of patients with RCC in the UK are being diagnosed incidentally, with almost three-quarters of these (74%) during investigation of symptoms unrelated to RCC. By contrast, a Norwegian study of 413 patients diagnosed with RCC between 1997 and 2010 reported a 53% incidental diagnosis rate, detected in 63% of these patients during follow-up for a pre-existing condition.[16] The reason for this difference is not certain but may reflect the different time periods under study, given the more liberal use of cross-sectional imaging over time.[17] Consistent with other studies, patients with an incidentally detected RCC tended to have smaller, lower stage and grade tumours than those presenting with related symptoms, but, nevertheless, almost one in five of patients identified incidentally had stage III/IV

disease at diagnosis. Whether patients who are diagnosed incidentally have better outcomes and potentially, therefore, different tumour biology, than those presenting with symptoms has been a matter of debate in the literature.[10 18–20] We did not find any difference in MFS or CSS between these two groups when matched for stage of disease, suggesting that incidental detection of advanced stage disease is not advantageous in terms of outcome.

Diagnosing kidney cancer early is, therefore, a significant public health challenge. Data from the 2010 National Cancer Patient Experience Survey in England report that almost 30% of 564 patients with renal cancer saw their general practitioner three or more times before hospital referral.[21] Furthermore, the results from the charity Kidney Cancer UK (KCUK) 2018 patient survey showed that 22% of the 153 responders who presented to their General Practitioner or an Accident & Emergency department waited more than 3 months for a diagnosis.[22] The results of the KCUK survey (n=175 in total) extend further, with 51% of patients reporting their cancer being detected incidentally during imaging for an unrelated reason, and less than one-third (31%) having symptoms due to RCC, reflecting the findings from our own, much larger, study.

How then do we improve the rates of early diagnosis in kidney cancer? Raising awareness among the public to present early to their doctor, even with vague symptoms may seem logical, as well as increasing awareness with primary care teams. But many patients remain asymptomatic until they have advanced stage disease, and the PPVs for symptoms other than haematuria, such as pain and fatigue, are even lower than 1%,[9] placing an impossible demand on general practitioners, who are required to act as gatekeepers to secondary care. The 5-year survival rates for kidney cancer in the UK lag behind the European average which may be related to differences in stage at diagnosis.[23] Greater availability of point-of-care ultrasound may make a significant impact but its use varies widely across Europe and has not been widely adopted in the UK, with potential barriers in terms of time and training.[24]

Interest in exploring the potential for kidney cancer screening is growing,[8 25] particularly given the significant predicted rise in incidence.[2] The potential cost-effectiveness of performing a single, renal focused, ultrasound scan among asymptomatic 60-year-old men has recently been reported.[26] However, numerous uncertainties still exist, in terms of who to screen, with what modality, as well as unknowns in terms of associated harms vs benefit.[27] This is an area that clearly warrants further research. The identification of robust diagnostic biomarkers either in the serum or urine of patients that could be used to easily rule in or out the presence of RCC is another priority area for study,[28] with recent promising reports in the literature,[29] although still requiring significant further validation and improved performance.

The strengths of this study include its prospective multi-centre design, among a contemporary cohort of patients with robust linked clinicopathological and outcome data. The eligibility criteria for the study were broad and we believe our patient cohort to be largely representative, when considered at a population level (for comparisons by age, sex, stage and RCC type see online supplementary table 2). It is possible that the proportion of patients in our study with stage IV disease may be slightly lower than in the true population, reflecting differences in the clinical pathway these patients may take, which may have impacted on our reported rate of incidental diagnosis. Furthermore, not all patients seen at participating centres with suspected RCC during the study period were recruited to the study and, overall, we acknowledge that our cohort size reflects only a small proportion (less than 10%) of all patients diagnosed with RCC in the UK during the study period. A further limitation is the fact that patient-reported symptoms were recorded following referral to secondary care and there may, therefore, be some element of recall bias.

In summary, this study draws attention to the fact that reliance on symptoms for the early detection of kidney cancer is not robust. Our data suggest that improving public and professional awareness will have only a limited impact, and innovative biomarkers for this purpose remain to be identified. We suggest it is time to re-examine the case for screening looking at opportunities to link RCC screening into other programmes such as low dose CT scans for lung cancer health checks or ultrasound-based screening for abdominal aortic aneurysms.

**Author affiliations**
[1]Leeds Institute of Medical Research at St James's, University of Leeds, Leeds, UK
[2]University of Cambridge, Cambridge, UK
[3]Department of Urology, NHS Lothian, Edinburgh, UK
[4]Department of Urology, Stockport NHS Foundation Trust, Stockport, UK
[5]Department of Urology, Saint James's University Hospital, Leeds, UK
[6]Department of Urology, University Hospital of Wales Healthcare NHS Trust, Cardiff, UK
[7]Department of Urology, Lister Hospital, Stevenage, UK
[8]Department of Urology, Charing Cross Hospital, London, UK
[9]Department of Urology, Queen Elizabeth University Hospital, Glasgow, UK
[10]Division of Cancer and Stem Cells, University of Nottingham, Nottingham, UK
[11]Department of Urology, Newcastle Upon Tyne Hospitals NHS Foundation Trust, Newcastle Upon Tyne, UK
[12]Department of Urology, Oxford University Hospitals NHS Foundation Trust, Oxford, UK
[13]Department of Urology, Northwick Park Hospital, Harrow, UK

**Acknowledgements** We are grateful to the patients and the staff at participating centres who participated in patient recruitment and data collection, within the Leeds Biobanking and Sample Processing Lab and at the Leeds Clinical Trials Research Unit.

**Contributors** Funding: PJS and REB; Study design and concept: PJS, REB, NSV; Patient recruitment and data collection: AA, JC, MK, SD, DHa, DHr, GO, PP, NS, GDS, MS, JW, NSV, REB; Data analysis: MW, GDS, NSV, REB; Manuscript preparation: NSV, MW, GDS, PJS, REB. All authors participated in the interpretation of results, critically revised the paper and approved the final version to be published.

**Funding** This project was funded by the National Institute for Health Research Programme Grants for Applied Research programme (project number NIHR PGfAR RP-PG-0707–10101) and has been published in full in Programme Grants for Applied Research Volume: 6, Issue: 3. Further information is available at: https://

www.journalslibrary.nihr.ac.uk/pgfar/pgfar06030/#/abstract. Additional support was provided by the NIHR Leeds Clinical Research Facility. The views and opinions expressed by the authors in this publication are those of the authors and do not necessarily reflect those of the NHS, the NIHR or the Department of Health.

**Disclaimer** The funder had no role in study design, data collection, data analysis or manuscript preparation.

**Competing interests** NSV discloses paid consultancy work and/or speaker honoraria for EUSA pharma, Bristol Myers Squibb, Pfizer, Merck and Novartis. GDS discloses paid consultancy work for Pfizer, EUSA Pharma, Merck, and CMR Surgical and an advisory board role for the latter. The other authors declare no conflicts of interest.

**Patient consent for publication** Not required.

**Ethics approval** The study was approved by the Leeds East Research Ethics Committee (ethical approval 10/H1306/6).

**Provenance and peer review** Not commissioned; externally peer reviewed.

**Data availability statement** All data relevant to the study are included in the article or uploaded as supplementary information.

**Author note** GDS's current affiliation is University of Cambridge, Cambridgeshire, UK

**ORCID iD**
Naveen S Vasudev http://orcid.org/0000-0001-8470-7481

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
