## [Reviewer comments · BMJ Open]

ARTICLE DETAILS

TITLE (PROVISIONAL)	The challenge of early renal cancer detection: symptom patterns and incidental diagnosis rate in a multicentre prospective UK cohort of patients presenting with suspected renal cancer
AUTHORS	Vasudev, Naveen; Wilson, Michelle; Stewart, Grant; Adeyoju, Adebajji; Cartledge, Jon; Kimuli, Michael; Datta, Shibendra; Hanbury, Damian; Hrouda, David; Oades, Grenville; Patel, Poulam; Soomro, Naeem; Sullivan, Mark; Webster, Jeff; Selby, Peter; Banks, Rosamonde

VERSION 1 – REVIEW

REVIEWER	Mette Nørgaard Aarhus University Hospital, Aarhus, Denmark
REVIEW RETURNED	23-Dec-2019

GENERAL COMMENTS	The current study seems to be a spin-off of a multicentre observational cohort study (reference 6 in the manuscript) including 608 patients with newly diagnosed RCC and 54 patients with a newly diagnosed benign renal mass. The objectives was to describe the frequency and nature of symptoms in patients presenting with suspected renal cell carcinoma and examine their reliability in achieving early diagnosis. This is a relevant aim, however, to study suspected renal cell carcinoma, it is important to include all patients suspected for RCC in a well-defined area and not only patents who ends up having an RCC diagnosis. It is not clear from the current description whether the study actually includes all patients with suspected RCC in the relevant period. This should be clarified. The design does not allow for assessment of the various symptoms reliability in achieving early diagnosis. This would require the estimation of e.g positive predictive value. The study population also includes patients who were not diagnosed based on any symptoms (i.e were incidental) and the study addresses the proportion of RCCs diagnosed based on incidental findings and also includes a prognostic part assessing whether incidental tumors have better prognosis. If this was intended to be part of the objectives for this study, it should be clearly list ed as study aims. In the current version of this manuscript the study aims and the study analyses do not fit well. It would be relevant to better describe the data – how large a proportion of patients suspected of RCC in UK were included in this study ?
--

	As stage was unevenly distributed between symptomatic patients and those with incidental findings it does not really make sense to compare survival overall. Only the survival curves stratified by stage are useful. In addition, a sentence like “symptomatic presentation was associated with poorer outcomes” (in the abstract) should be changed to reflect that this was likely explained by differences in stage distribution.
--	---

REVIEWER	Alejandro Sanchez Massachusetts General Hospital
REVIEW RETURNED	06-Jan-2020

GENERAL COMMENTS	The authors present a detailed analysis of symptoms associated with the presentation of renal cell carcinoma. Importantly they note that a high proportion of patients with metastatic disease do not present with symptoms and many patients with benign disease can present with concerning local/systemic symptoms. These results have implications for RCC screening. Comments:  - Can the authors comment on differences in symptoms by RCC histology groups (e.g., clear cell vs non-clear cell RCC). Comparisons within small cohorts in the non-clear cell group may be limited by small sample size. - Can the authors explain why patients with non-RCC related symptoms are not considered "incidental"? These patients were likely worked-up for some symptoms, e.g. abdominal pain, and diagnosed incidentally with a renal mass. - Would recommend removing the non-RCC symptoms from the survival analyses as these behave like the "incidental" group.
---

VERSION 1 – AUTHOR RESPONSE

Reviewer(s)' Comments to Author:

Reviewer: 1

The current study seems to be a spin-off of a multicentre observational cohort study (reference 6 in the manuscript) including 608 patients with newly diagnosed RCC and 54 patients with a newly diagnosed benign renal mass. The objectives was to describe the frequency and nature of symptoms in patients presenting with suspected renal cell carcinoma and examine their reliability in achieving early diagnosis. This is a relevant aim, however, to study suspected renal cell carcinoma, it is important to include all patients suspected for RCC in a well-defined area and not only patents who ends up having an RCC diagnosis. It is not clear from the current description whether the study actually includes all patients with suspected RCC in the relevant period. This should be clarified. The design does not allow for assessment of the various symptoms reliability in achieving early diagnosis. This would require the estimation of e.g positive predictive value.

- It is correct that we did not conduct a population-based study, as described by the reviewer, and we have deliberately avoided the use of this term to make this clear, but we have also now discussed this as a limitation. Our study comprised a prospective cohort of patients who were recruited following identification of a renal mass, that was very likely to, but not definitively, represent a renal cell carcinoma. We have made this clearer in the methods section. Amongst this defined cohort of patients we collected detailed information on their symptoms at the time of presentation, with further free-text annotation around the nature of the diagnosis, alongside very robust clinicopathological and

relapse/survival outcome data. To do so on a population level would be challenging and very costly. We wished to understand how the presence or absence and nature of symptoms relate to the characteristics of the newly diagnosed cancer (such as size, grade, stage, histology), how often the cancer is being diagnosed incidentally and whether reliance on symptoms is an effective strategy for achieving early diagnosis. Although not the focus of the study, we also present data on symptom patterns in those patients subsequently found to have a benign renal mass, to examine overlap with those presenting with RCC. We believe our findings are important in highlighting that, by the time symptoms are present, RCCs are often at an advanced stage and support the case for screening strategies to be urgently explored in this cancer. Our prospective cohort study complements the work of others who, for example, using retrospective review of UK GP records, describe the positive predictive value of symptoms such as haematuria, and which we reference in our manuscript (Shephard et al Br J Gen Pract 2013).

The study population also includes patients who were not diagnosed based on any symptoms (i.e. were incidental) and the study addresses the proportion of RCCs diagnosed based on incidental findings and also includes a prognostic part assessing whether incidental tumors have better prognosis. If this was intended to be part of the objectives for this study, it should be clearly listed as study aims. In the current version of this manuscript the study aims and the study analyses do not fit well.

- We agree and have now amended the manuscript to make the study aims clearer and more aligned with the study analyses.

It would be relevant to better describe the data – how large a proportion of patients suspected of RCC in UK were included in this study ?

- Our cohort represents only a small proportion of the total number of patients diagnosed with RCC in the UK per annum (less than 10%). This was a prospective study, consenting patients at the time of diagnosis and before intervention, across 11 UK centres, and with up to 5-year follow-up, with full details provided already in reference 6 of the paper. The eligibility criteria were broad, with inclusion of any patient with suspected RCC (except those with familial RCC, renal cancer acquired following/during renal replacement therapy and those at high-risk or with known blood-borne infectious disease). The clinical and tumour characteristics of the cohort are wholly in line with that expected and we believe are representative of the UK RCC population. Nevertheless, as above, we have now acknowledged this issue as a limitation of the study, both in the discussion section and amongst the key bullet points of strengths and limitations of the paper

As stage was unevenly distributed between symptomatic patients and those with incidental findings it does not really make sense to compare survival overall. Only the survival curves stratified by stage are useful. In addition, a sentence like “symptomatic presentation was associated with poorer outcomes” (in the abstract) should be changed to reflect that this was likely explained by differences in stage distribution.

- We agree and have amended the wording of the abstract to make this clearer. We feel it is reasonable to first present the data for all patients (incidental vs non-incidental diagnosis) (Fig 2A and D), before considering by stage. ie to demonstrate to the reader that a survival difference is present and that this can be explained by differences in TNM stage

Reviewer: 2

Comments:

Can the authors comment on differences in symptoms by RCC histology groups (e.g., clear cell vs non-clear cell RCC). Comparisons within small cohorts in the non-clear cell group may be limited by small sample size.

- We agree this is interesting to look at and have now done so. Few differences were found. Haematuria was present at the time of diagnosis in 65.2%, 52.5% and 40.6% of patients reporting local symptoms with chromophobe, clear cell and papillary cell carcinomas, respectively. However, even this difference was not statistically significant but may be limited by the small numbers, as highlighted. We have now included a line in the manuscript alluding to this.

Can the authors explain why patients with non-RCC related symptoms are not considered "incidental"? These patients were likely worked-up for some symptoms, e.g. abdominal pain, and diagnosed incidentally with a renal mass.

- Both patients who were recorded as having no RCC type symptoms and those with RCC-type symptoms considered unrelated, were included amongst the 351 incidentally diagnosed patients. We have now tried to further clarify this in the text (methods section).

Would recommend removing the non-RCC symptoms from the survival analyses as these behave like the "incidental" group.

- We presume this point relates to the misunderstanding above regarding which patients comprised the incidentally diagnosed group of patients.

VERSION 2 – REVIEW

REVIEWER	Mette Nørgaard Aarhus University Hospital, Denmark
REVIEW RETURNED	05-Feb-2020

GENERAL COMMENTS	The authors have responded nicely to the comments. However, they have not included much of this information in the revised manuscript and therefore the value of the manuscript remain unclear. First of all, it is difficult as a reader of the manuscript to understand the selection process into the study. This is a multicentre study including patients with a renal mass suspicious of RCC. Thus, the tumor was very likely to represent a RCC but this was not confirmed. Does this mean that imaging tests were performed but no biopsies were taken at time of study inclusion, or? This could be better described in the manuscript. It would be really useful to present a flowchart – if possible – to describe the selection process of study participants. Also, it should be clear how many centers contributed patients and whether these centers each included all their potential RCC patients during the study period. The authors emphasize that this is a prospective cohort study. However, focus is on presenting symptoms at time of diagnosis, which the patients were asked about, and, as the authors also clearly state, recall bias cannot be excluded. Accordingly, some of the data are retrospective in nature. Therefore, the emphasis on
---

	the prospective nature should be moderated and better discussed. Why is it a specific problem that previous studies estimating the prevalence of incidental diagnoses were retrospective in nature? The selection process into the present study may differ from the selection into studies in the existing literature. Such differences could lead to differences in the prevalence of incidental RCCs—this should also be discussed. The authors argue that they believe their patient cohort is representative. It is not clear what it is representative of – but most likely the UK RCC population? Would it be possible to present e.g the age, sex, stage distribution of the overall RCC population in UK? A comparison between patient characteristics of the study population and those in the general RCC population could help argue why it is likely representative.
--	--

REVIEWER	Alejandro Sanchez, MD Massachusetts General Hospital
REVIEW RETURNED	21-Jan-2020

GENERAL COMMENTS	The authors address all of my initial concerns.
---

VERSION 2 – AUTHOR RESPONSE

Reviewer(s)' Comments to Author:

Reviewer: 2

Please leave your comments for the authors below

The authors address all of my initial concerns.

We are pleased to note that Reviewer 2 feels that their initial concerns have been satisfactorily addressed

Reviewer: 1

We thank Reviewer 1 for their further helpful comments. We have tried to carefully and thoroughly address these, both here and through addition to the manuscript itself.

The authors have responded nicely to the comments. However, they have not included much of this information in the revised manuscript and therefore the value of the manuscript remain unclear.

We apologise for this and have made a number of additional changes in the manuscript, as below, to address this

First of all, it is difficult as a reader of the manuscript to understand the selection process into the study. This is a multicentre study including patients with a renal mass suspicious of RCC. Thus, the tumor was very likely to represent a RCC but this was not confirmed. Does this mean that imaging tests were performed but no biopsies were taken at time of study inclusion, or? This could be better described in the manuscript. It would be really useful to present a flowchart – if possible – to describe the selection process of study participants.

We think the addition of a flow diagram to explain the selection process is an excellent idea and we have now incorporated this as a main figure (Fig 1). This should help to clarify to the reader that patients were approached and consented to take part in the study before the diagnosis of RCC was confirmed, following subsequent surgery or biopsy. This design was employed to allow collection of research blood samples pre-operatively, early capture of presenting symptoms and recruitment of patients with benign renal masses for comparative purposes. As well as the addition of a flowchart, we have also now made it clearer in the text that patients were approached to take part in the study prior to surgery or biopsy.

Also, it should be clear how many centers contributed patients and whether these centers each included all their potential RCC patients during the study period.

In total, 11 UK centres took part in the study (stated in the first line of Results section). However, we have now also included, as a supplementary table (Supplementary Table 1), the total number of patients that each centre recruited, the time over which these patients were recruited and the derived recruitment rate (patients/month). As in all such studies, the time at which each study opened to recruitment varied (due to differences in local set up times) and, hence, some centres recruited over a longer period than others. Centres also varied in terms of their patient volume as well as capacity to recruit to the study. Given its nature, it is not possible that all potential RCC patients were recruited during the study period from any one centre. We now also acknowledge this, alongside the fact that our cohort represents only a small proportion of the total newly diagnosed UK RCC population.

The authors emphasize that this is a prospective cohort study. However, focus is on presenting symptoms at time of diagnosis, which the patients were asked about, and, as the authors also clearly state, recall bias cannot be excluded. Accordingly, some of the data are retrospective in nature. Therefore, the emphasis on the prospective nature should be moderated and better discussed. Why is it a specific problem that previous studies estimating the prevalence of incidental diagnoses were retrospective in nature?

We agree that this should be better discussed and we now do this on page 11 of the manuscript in the Discussion section. We consider the recording of patients' symptoms to be prospective in nature in our study since these data and details regarding the diagnosis were recorded in a planned and specific way using standardised case report forms, for the purpose of later examining symptom patterns and relating them to tumour characteristics and patient outcomes. Symptoms were also recorded as early as possible following referral of patients to secondary care, minimising issues with recall bias. Our study design is different to other studies that have retrospectively looked back at patient records and that, whilst often feasible on a much bigger scale, may suffer from things such as presenting symptoms not always being accurately or wholly recorded and with a lack of sufficient detail when trying to determine whether incidental in nature or not.

The selection process into the present study may differ from the selection into studies in the existing literature. Such differences could lead to differences in the prevalence of incidental RCCs– this should also be discussed.

The selection criteria were very broad in our study – any patient with a suspected kidney cancer was eligible, regardless of stage. As such, the selection of patients in our study was similar to other previously reported studies reporting incidental diagnosis rates amongst newly diagnosed RCC patients. However, given that patients were principally recruited to our study by urologists, it is possible that some patients presenting with stage IV disease were not seen via this route, possibly going direct to oncology. This relates to the reviewer's comment below and we acknowledge that the proportion of stage IV patients may be lower in our cohort than in the true population. We have now added comment on this in the limitations section of the Discussion.

The authors argue that they believe their patient cohort is representative. It is not clear what it is representative of – but most likely the UK RCC population?

Would it be possible to present e.g the age, sex, stage distribution of the overall RCC population in UK? A comparison between patient characteristics of the study population and those in the general RCC population could help argue why it is likely representative.

We fully agree that this is an important and useful thing to demonstrate and we have now created a comparative table to illustrate this (Supplementary Table 2). Complete and accurate population-based figures for the UK from a single source are in fact difficult to find. Stage at diagnosis of kidney cancer, for example, from 2017, is missing in 13% of patients. For completeness, we have included population data from the US, as well as data from a global study involving >4000 patients. The data support the representative nature of our cohort in terms of age, sex, stage and tumour type. The possible exception is in terms of the percentage of patients presenting with stage IV disease (9% (current) vs 20% (UK) vs 15.8% (US)). The 20% figure may, however, be an overestimate, since we would predict that many of those patients with missing data would have localised disease. Furthermore, a general stage shift in RCC presentation is also well documented, with a recent report from the US suggesting that only 11% of patients now present with stage IV disease (Patel et al. Clinical Stage Migration and Survival for Renal Cell Carcinoma in the United States. *Eur Urol Oncol.* 2019). Nevertheless, given the uncertainty and the fact that this may impact on our incidental

diagnosis rate, we have acknowledged this issue as a potential limitation of the study, as described above

VERSION 3 – REVIEW

REVIEWER	Mette Nørgaard Aarhus University Hospital, Denmark
REVIEW RETURNED	21-Feb-2020
GENERAL COMMENTS	The manuscript has improved by the latest additions. I have no further comments